# Deep Interest Network Based on Knowledge Graph Embedding

**Dehai Zhang** * , **Haoxing Wang, Xiaobo Yang, Yu Ma** , **Jiashu Liang and Anquan Ren**

School of Software, Yunnan University, Kunming 650504, China
* Correspondence: dhzhang@ynu.edu.cn

**Abstract:** Recommendation systems based on knowledge graphs often obtain user preferences through the user's click matrix. However, the click matrix represents static data and cannot represent the dynamic preferences of users over time. Therefore, we propose DINK, a knowledge graph-based deep interest exploration network, to extract users' dynamic interests. DINK can be divided into a knowledge graph embedding layer, an interest exploration layer, and a recommendation layer. The embedding layer expands the receptive field of the user's click sequence through the knowledge graph, the interest exploration layer combines the GRU and the attention mechanism to explore the user's dynamic interest, and the recommendation layer completes the prediction task. We demonstrate the effectiveness of DINK by conducting extensive experiments on three public datasets.

**Keywords:** recommendation; knowledge graph embedding; interest explore

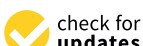



## 1. Introduction

Due to the diversity and heterogeneity that information knowledge graphs represent, they are widely used in high-level tasks of deep learning, such as image captioning and text generation. These methods follow a unified paradigm that uses representation learning methods to map high-dimensional sparse features and structured features of knowledge graphs into a vector space and learn the relationship between the vectors through the model. These methods utilize the fitting ability of deep neural networks to automatically learn features and improve the generalization ability of the model. The relevant researchers of recommender systems have found in continuous practice that the user's interest preference is a dynamic process. As shown in Figure 1, a user has a soft spot for Apple's electronic products, even if he also has other products at the same time. In other words, the preferences for different items are not the same in the user's historical interaction sequence. To extract user sequence interaction features, researchers propose time-series recommendation methods to capture users' dynamic interests. These methods draw on the successful experience in the field of natural language processing and actively explore how to design recurrent neural networks to extract users' long-term and short-term interest preference features.

This method has effectively solved the problems of data sparseness and cold start in the recommendation, but it still has the following shortcomings:

(1) Recommendation methods based on knowledge graphs usually learn the interaction characteristics between users and items in a static click matrix, ignoring that the interactions between users and items are dynamic and orderly. At the same time, these models are only suitable for offline training and cannot achieve online training and real-time recommendation.

(2) Most of the current time series recommendation methods bluntly "migrate" the models in the field of natural language processing to the recommendation task. However, in the recommendation task, the user's interaction records are often not closely continuous, i.e., the user's interaction sequence may be jumping time-series data across time and platforms. It is difficult for models in the traditional natural language processing field to learn the time-series features in jumping interaction sequences. Most of the current time

series recommendation methods do not introduce knowledge graphs to assist in exploring the diverse interests and preferences of users at different times.

We propose DINK (deep interest network based on knowledge graph embedding) to overcome these shortcomings. The main work of this paper is as follows:

(1) DINK maps the click sequence to a click sequence path in the knowledge graph, and expands the user preference receptive field at different times through the aggregator.

(2) DINK combines attention with GRU to explore users' dynamic interests and infer their interest at the next moment.

(3) We conduct CTR prediction experiments, Top-k recommendation experiments and performance experiments on three public datasets. Compared to other models, our experimental results show that DINK can more effectively extract the dynamic interest features of users and can effectively alleviate the cold start problem.

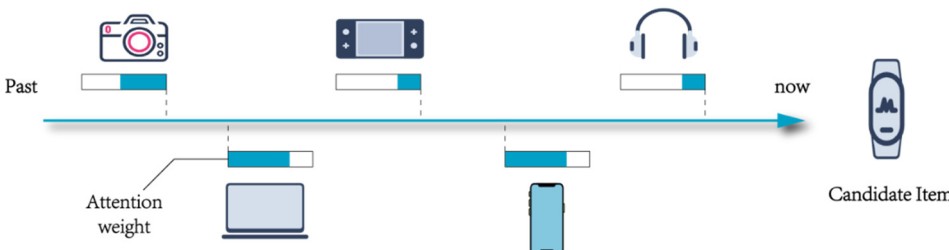

**Figure 1.** Illustration of dynamic interests.

## 2. Related Work

### 2.1. Recommendation Systems

Currently, researchers divide the recommendation algorithms [1] based on knowledge graph into three categories. (1) Recommendation algorithm based on knowledge graph embedding [2]. (2) Recommendation algorithm based on knowledge graph path [3]. (3) HyBrid, which is a combination of the two [4].

The Recommendation algorithm based on knowledge graph Embedding represents entities and relationships by the method of graph embedding, and then expands the semantic information of the original item and user representation. DKN [5] (deep knowledge-aware network) introduces the knowledge graph into the news recommendation task. It takes a candidate news and a user's click history as input, pretrains the entities in the knowledge graph, uses the VGG network [6] for feature extraction, and the output is the probability of the user clicking the news. However, this method only uses the words in the title to make a recommendation and does not effectively utilize the more valuable context information. CKE [7] (collaborative knowledge base embedding for recommender) uses three embedding methods to fit the knowledge graph, text, and visual information in the knowledge base. Then, it combines the three vectors and the implicit feedback of users for the recommendation system.

The relationship between nodes and paths in knowledge graphs is usually many-to-many. The information contained in these paths can often be used as a supplement to enhance the effectiveness and interpretability of the recommendation system. KPRN [8] (explainable reasoning knowledge graph recommendation) combines the semantics of entities and relations to generate path representations and uses LSTM [9] (long short term memory) to effectively reason about the path by using sequential dependencies in the path, and then infer the rationale of user item interaction. At the same time, a new weight pool operation is designed to distinguish the importance of different paths in connecting users and items, so that the model has a certain degree of interpretability. The model using path traversal can obtain full interpretability and a certain performance improvement, but it often consumes a lot of computing resources when learning the optimal path. Rule Guidance [10] proposed a path-based rule mining method, which introduces reinforcement learning [11] into relational agents for learning. High-quality rules generated by symbol-based methods are utilized to provide reward supervision for the swing-based agent. This

method improves the performance of walk-based models without losing interpretability, but it does not perform well on large-scale datasets.

HyBrid refers to recommendation algorithms that combine graph paths with knowledge graph embeddings, and knowledge graph recommendation algorithms based on graph neural networks are also classified in this category. KGCN [12] (knowledge graph convolutional networks for recommender systems) captures the correlation between items by sampling from the neighbors of each entity in the knowledge graph and mining the associated attributes of items on the knowledge graph. The user's potential interests are captured at a distance. NACF [13] (neighborhood aggregation collaborative filtering based on knowledge graph) iteratively encodes the potential information of the knowledge graph into user features and uses the attention mechanism in GCN [14] (graph convolutional network) to consider the personalized preferences and multiple associations of users during information aggregation.

### 2.2. Knowledge Graph Embedding

Knowledge graph embedding [15] refers to the operation of representing entities and relationships in a knowledge graph by vectors. The vector representations of $(h, r, t)$ triples are obtained by training each other to facilitate the sharing of features in the knowledge graph. Specific training methods can be divided into two broad categories:

(1) Translation distance models, including TransE, TransH, TransR [16,17], and other varieties. They regard the Tail vector as the translation distance obtained from the Head vector through the Relation vector, and the scoring function can be regarded as the Euclidean distance between vectors.

(2) Semantic matching models [18,19] score by calculating the semantic similarity between the Head vector and the Tail vector after the linear transformation of the Relation space. The score function can be considered as the angle between vectors. Through the click exposure logs of query and document, DSSM [20] expressed query and document as low-dimensional semantic vectors, calculated the distance of two semantic vectors by cosine similarity, and finally trained the semantic similarity model. DistMult [21] simplifies the relationship matrix to a diagonal matrix, which has the advantage of being extremely efficient, but it is oversimplified and can only deal with symmetric relationships, which cannot be fully applied to all scenarios.

### 2.3. GRU

A sequence is a set of data streams that have pre-and post-dependencies, and the position changes of data elements in the whole sequence will affect the features expressed in the whole sequence. Researchers proposed a cyclic neural network to extract the features of the sequence. At present, common recurrent neural networks include RNN [22], LSTM, and GRU [23].

RNN is a sequential recursive deep neural network that has achieved good results in many sequential tasks. RNN extracts the sequence features of elements by setting the hidden layer $h_t$ of the n-layer stack, and each hidden layer extracts the sequence features of different moments. Due to the features of the iterative learning sequence of RNN, its gradient accumulates continuously with the length of the sequence, eventually leading to gradient explosion (gradient continuously accumulates toward positive infinity) or gradient disappearance (gradient continuously accumulates toward 0), thus losing the long-term features of the sequence. LSTM improves the hidden layer in RNN, in which forgetting gate $f_t$, update gate $i_t$, and output gate $o_t$ are set to filter and discard redundant features. GRU is an improvement of LSTM, which filters the information of the last moment and transmits the information of the current moment through a deep update gate. Since GRU only uses one parameter to control the transmission of information, the number of parameters is reduced by 25% compared with LSTM.

Much work has been done in the field of recommendation to extract the behavior characteristics of user click sequences by using recurrent neural networks, and good results

have been achieved. GRU4Rec [24] regarded the user's click sequence as a session, and after each item in the sequence went through N layers of GRU. The extracted features were used to predict the click summary of each item at the next moment. Bert4rec [25] introduced Bert into the recommendation task and designed a deep two-way attention to model the sequence of user behaviors, which allowed each project to integrate the information from the left and right sides to predict the recommendation generalization at the next moment. We combine GRU with an attention mechanism and propose a dynamic interest inference layer to capture the dynamic sequence features of users.

## 3. Our Approach

### 3.1. Framework

DINK consists of three parts: an embedding layer, an interest exploration layer and a prediction layer, and its framework is shown in Figure 2. In the embedding layer, the user interaction sequence is mapped into the recommended scene knowledge graph, and the features of the first-order neighbor nodes are extracted by the aggregator to obtain the central aggregation vector $(E(v_1), E(v_2) \ldots E(v_t))$. At the same time, user features and candidate item features are also initialized to obtain their embedding vectors. In the dynamic interest reasoning layer, DINK calculates the similarity score between the user interaction sequence and the candidate item as the attention weight and then uses the weighted center aggregation vector as the input of the GRU to obtain the user's dynamic interest feature. In the prediction layer, the user vector, the candidate item vector, and the dynamic interest vector are spliced and pooled to predict the score of the candidate item.

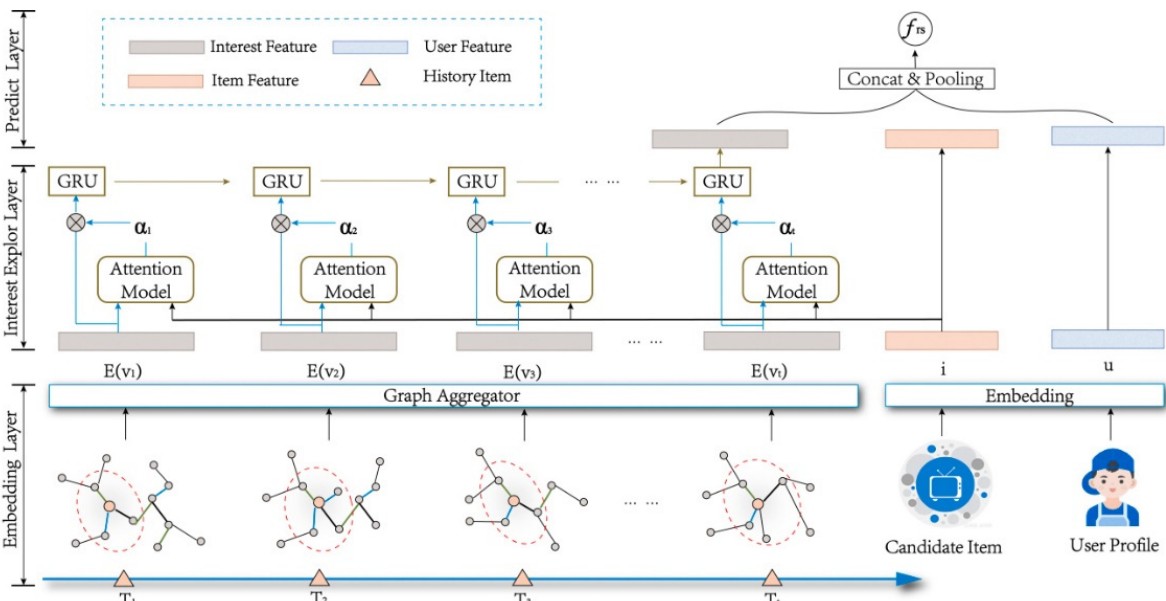

**Figure 2.** The Framework of DINK.

In DINK, a user set $U = \{u_1, u_2, \ldots, u_n\}$ with N users is defined, and item set $I = \{i_1, i_2, \ldots, i_m\}$ with M items and the user's historical interaction. The item set $V = (V_1, V_2 \ldots, V_u)$, where $V_i = (v_1, v_2, \cdots, v_t)$ represents the historical interaction records at T moments, and V is the set of historical interaction sequences of all users. At the same time, this section collects the corresponding scene knowledge graph for each dataset, which is defined as $G = \{(h, r, t) | h, r \subseteq E, r \subseteq R\}$, where $I \subseteq E$ represents the user click sequence. Items are fully contained within the knowledge graph entities.

### 3.2. Embedding Layer

The recommendation based on knowledge graph usually introduces the features in the knowledge graph into the recommendation task. It generally consists of three stages: (1) knowledge graph representation learning, (2) obtaining vector representation, (3) extracting vector features. DINK attempts a novel pattern that introduces a temporal dimension to item features in recommendation tasks. First, the user interaction sequence $\langle v_1, v_2, \ldots, v_t \rangle$ is mapped to the entity nodes in the knowledge graph. Then, the entity node is used as the center to extract the features of its surrounding neighbor nodes. Finally, the center aggregation vector is used to represent the user at this moment of diverse interests.

In the actual recommendation scene knowledge graph, an entity has multiple semantic relationships, and the number of relationships owned by different entities is inconsistent. To make the model more standardized and efficient, we collect the same number of neighbor nodes for each entity in the user interaction sequence to form the user interaction subgraph sequence $G(v_t) = (e_1, e_2, \ldots, e_k)$, where $G(v_t)$ represents the user interaction subgraph at time T, K represents the number of subgraph samples, and the complete user interaction subgraph sequence is represented as $\langle G(v_1), G(v_2), \ldots, G(v_t) \rangle$. Finally, DINK uses different aggregators to calculate the central aggregation vector of the user interaction subgraph. Equation (1) calculates the central aggregation vector $E(v_t)$ at time T. Agg is the aggregator, and the aggregation vector of the user interaction sequence is expressed as $\langle E(v_1), E(v_2), \ldots, E(v_t) \rangle$.

$$E(v_t) = Agg(G(v_t)) = Agg(e_1, e_2, \ldots, e_k) \tag{1}$$

To capture user preferences at different moments, this chapter proposes an SUM aggregator, a CONCAT aggregator, and an ATTENTION aggregator to compute the center aggregation vector.

SUM aggregator

The SUM aggregator adds the center node vector and the neighbor node vector, and then performs nonlinear transformation.

$$E(v_t) = \sigma \left( W \cdot \left( v_t + \sum_{e \subseteq G(v_t)} e \right) + b \right) \tag{2}$$

CONCAT aggregator

The CONCAT aggregator adds the neighbor node vectors, then concatenates them with the center node vector, and finally performs nonlinear transformation, as shown in Equation (3). $W, b \in \mathbb{R}^{2d}$ are learnable weights and biases, $\sigma$ is a nonlinear activation function, and $\|$ is a vector stitching operation.

$$E(v_t) = \sigma \left( W \cdot \left( v_t \parallel \sum_{e \subseteq G(v_t)} e \right) + b \right) \tag{3}$$

ATTENTION aggregator

The attention aggregator calculates the attention weights of neighbor nodes and nonlinearly transforms the weighted neighbor node vectors.

$$\alpha_{r,e} = softmax(\sigma(W_r \cdot (e_i \odot r_i))) \tag{4}$$

$$E(v_t) = \sigma \left( W \cdot \sum_{e \subseteq G(v_t)} \alpha_{r,e} \cdot e + b \right) \tag{5}$$

### 3.3. Interest Explore Layer

DINK adds an attention mechanism to the user interaction sequence, which considers the correlation between the central aggregation vector and candidate items at different times and calculates an attention weight for the central aggregation vector at all times.

$$\alpha_t = \frac{\exp(E(v_t)W_e)}{\sum_{j=1}^{T} \exp(E(v_j)W_e)} \tag{6}$$

After obtaining the weight of the user-division sequence, it will take advantage of the central aggregation vector to indicate the user's interest in different moments. To explore the dynamic evolution of interest, DINK uses GRU to extract users' long short-term interest preference characteristics.

$$z_l = \sigma(W_z x_l + U_z h_{l-1}) \tag{7}$$

$$r_l = \sigma(W_r x_l + U_r h_{l-1i}) \tag{8}$$

$$h_l = \tan h(W x_l + U(r_l \times h_{l-1})) \tag{9}$$

$$h_l = (1 - z_l) \times h_{l-1} + z_l \times h_l \tag{10}$$

where $l = (1, 2, \dots, t)$ represents the moment of the user sequence, and DINK retains the output vector $h_t$ at the last moment as the user's dynamic interest preference feature. $W_z, W_r, W \in \mathbb{R}^{2d}$ are the trainable weights and biases of each cell of the GRU.

### 3.4. Predicted Layer

To enrich the features for recommendation, DINK uses L-layer fully connected layers to extract high-level features of users and candidate items when predicting item ratings, stitches them together with dynamic interest vectors, and uses activation functions to calculate ratings. The process is shown in Equation (12). $W_p \in \mathbb{R}^{3d}, b_p \in \mathbb{R}^{3d}$ are trainable weights and biases to extract the features of the spliced vector.

$$i^L = M^L(i) \tag{11}$$

$$u^L = M^L(u) \tag{12}$$

$$\widehat{y}_{ui} = \text{sigmoid}\left(W_p\left(u^L \parallel i^L \parallel h_t\right) + b_p\right) \tag{13}$$

### 3.5. Training

DINK regards the recommendation task as a binary classification problem. The label is 1 if the user interacts with the item. Otherwise, the label is 0. At the same time, DINK adopts the pointwise approach to train the parameters of the model, and uses the cross entropy as the loss function.

$$\mathcal{L} = - \sum_{(u,i)\in\mathcal{O}^+} \log \widehat{y}_{ui} + \sum_{(u,i)\in\mathcal{O}^-} \log(1 - \widehat{y}_{ui}) + \lambda_1 \parallel \theta \parallel_2^2 \tag{14}$$

where $\mathcal{O}^+ = \{(u,i)|y_{ui} = 1\}$ represents a positive sample, and $\mathcal{O}^- = \{(u,i)|y_{ui} = 0\}$ represents a negative sample. Furthermore, to prevent overfitting, DINK also introduces L2 regularization to the training process to train $\theta$, which is a trainable weight in the layer.

The DINK framework is shown in Algorithm 1:

---

**Algorithm 1:** DINK algorithm.

---

➤　input: $U$: Set of user; $I$: Set of item; $G_{\langle E,R \rangle}$: Knowledge graph; $V$: User interaction sequence

➤　output:Prediction score $F(u,i \big| \theta, G_{\langle E,R \rangle})$

---

1　initialize $U, I, E, R, w$ as trainable parameters;
2　**while DINK** not converge **do**:
3　//Embedding
4　　Extracting subgraphs$\rightarrow V \rightarrow G(v_1), G(v_2), \dots, G(v_t)$;
5　　**for** $T$ in length ($V$) **do**:
6　　　$E(v_t) = Aggregator(G(v_t))$
7　　**end**
8　　//Interest Explore Layer
9　　**for** $E(v_t)$ in length ($V$) **do**:
10　　　$\alpha_t = \text{Attention}(E(v_t), i)$
11　　　$h_t = \text{LSTM}(\alpha_t E(v_t))$
12　　　**end**
13　　//Predicted
14　　　$\widehat{y}_{ui} = f(concat(h_t, u, i))$
15　　Updating training parameters;
16　**end**

---

　　The input of DINK includes a set of users, a set of items, a knowledge graph of recommended scenarios, and historical interaction sequences of users. Since the set of items is contained in the knowledge graph, the users and entities and relations in the knowledge graph meet the Gaussian distribution after vector initialization. In the embedding layer, the central aggregation vector is calculated by different aggregators for the interactive session. In the dynamic interest inference layer, the center aggregation vector calculates the attention weight of the candidate items, the weighted center vector is used as the input of GRU, and GRU outputs the dynamic interest vector of the user. In the recommendation layer, the user feature vector, the feature vector of the candidate item, and the dynamic interest vector of the user are concatenated, and the confidence of the recommendation is calculated by the scoring function. It can be seen from the above that the trainable parameters of DINK consist of aggregators and GRU units at different times, which requires a large number of parameters to learn. Therefore, the Adam algorithm is used to match adaptive learning rates for different parameters.

## 4. Experiment

### 4.1. Dataset

　　In this experiment, we verify the effectiveness of DINK on three public datasets, MovieLens-1M, Last.FM and Book-Crossings. In addition, this paper collects the corresponding scene knowledge graphs for the three datasets.

　　MovieLens-1M not only collects user click records and rating records, but also collects movie metadata and user attribute data. The metadata of the movie include the genre, style, and era of the movie, while the attribute data of the user include demographic information such as the user's age, gender, and occupation. MovieLens-1M is a subset of the MovieLens dataset, which contains approximately 1 million display ratings (ratings from 1 to 5) for 4000 movies by 6000 users. The knowledge graph corresponding to MovieLens contains 17,633 entities and 20,195 relationships.

　　Last.FM collects rating data from users of the Last.FM online music platform. This dataset collects contextual information about music data, including each user's favorite artist, song list, and number of plays. In addition, Last.FM also collects user demographic information including user age and occupation. Last.FM dataset contains 100,000 display ratings data (ratings from 1 to 352,698) of 2000 users. The knowledge graph corresponding to Last.FM contains 9366 entities, 15,518 relations, and 60 relation types.

Book-Crossings contains nearly 1.1 million ratings (ratings ranging from 1 to 10) on 270,000 books by 90,000 users, including both explicit and implicit ratings. The scene knowledge graph corresponding to Book-Crossings contains 12,431 entities and 19,793 relationships.

The details of these three datasets are shown in Table 1

**Table 1.** Basic statistics of the dataset.

|  | MovieLens-1M | Last.FM | Book-Crossings |
|---|---|---|---|
| #Users | 6036 | 1872 | 17,860 |
| #Items | 2347 | 3846 | 14,910 |
| #Ratings | 753,772 | 423,461 | 139,746 |
| #Rating Type | 3 | 10 | 5 |
| #Density of Dataset | $2.8 \times 10^{-3}$ | $58.8 \times 10^{-3}$ | $0.5 \times 10^{-3}$ |
| #Triple | 20,195 | 15,518 | 19,793 |
| #Density of Graph | $9.49 \times 10^{-5}$ | $0.35 \times 10^{-5}$ | $4.99 \times 10^{-5}$ |

*4.2. Baseline*

LibFM [26] is a machine learning method that decomposes matrix features through gradient descent and Bayesian inference. It is suitable for high-order sparse matrices. It is a classic decomposition model method. Due to its simple structure and remarkable effect, it is widely used in industry.

Wide&Deep [27] integrates the deep model and the shallow wide model for training, effectively combining the memory ability of the shallow model and the generalization ability of the deep model, which is an innovation in the field of engineering.

CKE [7] proposes a knowledge graph coordination filtering framework, which uses a unified recommendation framework to jointly embed multimodal data, such as knowledge graph, text information, and image information, into recommendation tasks. It is the SOTA model in the field of knowledge graph recommendation in 2016.

RippleNet [28] was published in CIKM in 2018. This work proposes a hybrid method that uses knowledge graph structure information to assist recommendation, which completes personalized recommendation tasks by exploring users' potential interest features on the knowledge graph.

KGCN was published on WWW in 2019. This work proposes a recommendation model that uses graph convolutional network to extract knowledge graph features.

MKR [29] was published in KDD in 2020. This work proposes a multi-task feature learning method that combines knowledge embedding and recommendation tasks. By combining knowledge graph embedding algorithms and recommendation system modules, the potential information of recommendation scenarios and knowledge graphs can interact with each other.

GMCF [30] was published in SIGIR in 2021. This work proposes a collaborative filtering model based on neural network graph matching, which divides the knowledge graph into multiple subgraphs, and designs a matching algorithm to model and aggregate attribute interactions in the graph structure, effectively extracting the internal, multi-level features between subgraphs.

*4.3. Metrics*

The AUC and ACC are widely used metric in CTR prediction. It measures the goodness of order by ranking all the ads with predicted CTR, including intra-user and inter-user orders.

$$ACC = \frac{TP + TN}{TP + TN + FP + FN} \tag{15}$$

$$AUC = \frac{\sum I(P_{pos} + P_{neg})}{M * N} \tag{16}$$

In the ACC formula, TP is a predicted correct positive sample number, and FP is a positive sample number of predictive errors. TN is the number of predicted correct negative

samples, and FN is a negative sample number of predictive errors. In the formula of the AUC, m and n represent the number of positive samples and negative samples in the dataset, respectively, and M ∗ N represents the number of sample pairs. The number of predicted probabilities of the positive sample in the sample is greater than the number of predicted probabilities of the positive sample.

*4.4. Performance in CTR*

The result of the hit rate estimation experiment is shown in Table 2. Among them, DINK-K indicates that the user's interaction sequence does not introduce a knowledge graph. At this time, the input vector of the dynamic interesting layer is a random initialized user sequence vector. DINK-A indicates that the model does not introduce attention mechanisms in the dynamic interest in the reinforcement layer. At this time, the dynamic interest in the reinforcement layer is an ordinary GRU model whose input is a polymeric user sequence vector. (-ATT), (-SUM), and (-CAT) indicate the ATTENTION aggregator, SUM aggregator, and CONCAT aggregators, respectively. By analyzing the data analysis of Table 2, the following conclusions can be drawn:

- DINK is better in the case of data density and high knowledge density

The scene knowledge map density of MovieLens-1M and Last.fm is 8.04 and 4.03, respectively, the density of the Book-Crossing scene knowledge map is only 1.03, and the user's interaction data are lower. By observing the results, it is possible to find that the performance of DINK in these two datasets exceeds the performance in Book-Cross, whether from the absolute value or relative value. In the Book-Crossing AUC metrics, DINK is backward to the latest GMCF model 6%, while in two datasets in MovieLens-1M and Last.fm, DINK's performance exceeds the rest of the baseline model. Therefore, DINK can use external knowledge to enhance the recommended effect, but compared to the nearest excellent model, there is still no improvement in overcoming data sparseness. Attention aggregator tables in three datasets are due to other aggregators. This is because the attention aggregator fuses the weight information of the relationship edges in the aggregation process, which can effectively extract the features strongly related to the items.

- Compared to attention mechanisms, neighbor nodes in knowledge maps are more meaningful

In the three datasets, DINK-K grew more than DINK-A, which means that knowledge representation is a more key promotion of the semantic feature in the study. At the same time, it is noted that DINK-K and DINK-A decline in the Book-Crossing data concentration to a larger degree, indicating that under the sparse scene, the attention mechanism can better play the role, and the external information, such as the knowledge map, is also more important.

**Table 2.** Performance in CTR.

| Method | MovieLens-1M | | Last.FM | | Book-Crossing | |
|---|---|---|---|---|---|---|
| | AUC | ACC | AUC | ACC | AUC | ACC |
| Wide&Deep | 0.898 | 0.820 | 0.756 | 0.688 | 0.712 | 0.624 |
| CKE | 0.801 | 0.742 | 0.744 | 0.673 | 0.671 | 0.673 |
| Ripple | 0.920 | 0.842 | 0.780 | 0.702 | 0.729 | 0.662 |
| LibFM | 0.892 | 0.812 | 0.777 | 0.709 | 0.685 | 0.640 |
| MKR | 0.921 * | 0.847 * | 0.796 * | 0.739 * | 0.738 | 0.688 |
| KGCN | 0.917 | 0.837 | 0.796 | 0.728 | 0.731 | 0.678 |
| GMCF | 0.918 | 0.845 | 0.789 | 0.711 | 0.789 * | 0.712 * |
| DINK-ATT | **0.929** | **0.851** | **0.817** | **0.752** | **0.745** | **0.702** |
| DINK-SUM | 0.925 | 0.851 | 0.813 | 0.745 | 0.730 | 0.688 |
| DINK-CAT | 0.922 | 0.848 | 0.808 | 0.742 | 0.741 | 0.697 |
| DINK-K | 0.903 | 0.826 | 0.782 | 0.742 | 0718 | 0.668 |
| DINK-A | 0.921 | 0.839 | 0.808 | 0.746 | 0.726 | 0.679 |
| Improve (%) | 0.5% | 0.3% | 2.5% | 1.7% | −4.2% | −1% |

### 4.5. Performance in Cold Start Simulation

This section verifies whether DINK can effectively alleviate the cold-start problem. We simulate the cold-start environment by adjusting the proportion of the training set to increase the proportion of "new users". The AUC results of each model in the cold start environment are shown in Figure 3. This shows that DINK can perform better in a cold start environment. In addition, through further observation of the results, it can be found that the recommendation model based on a knowledge graph performs better than traditional recommendation models, such as Wide & Deep, which also proves that the introduction of a knowledge graph can effectively alleviate the cold start problem in recommendation tasks.

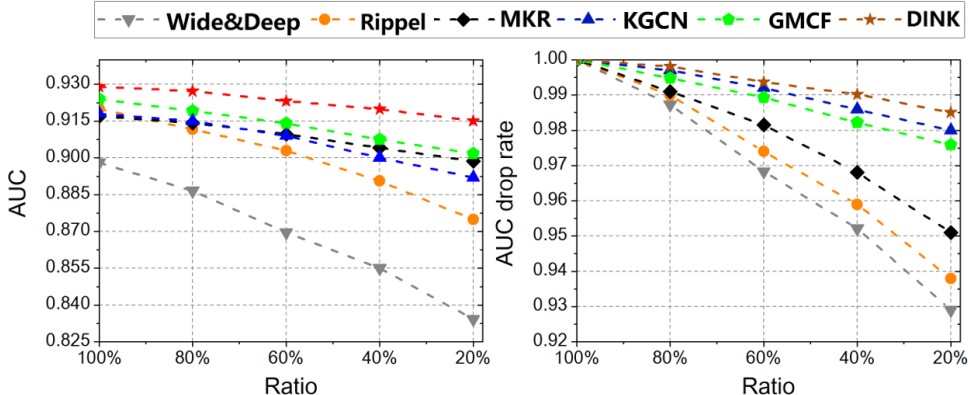

**Figure 3.** Performance of each model in the cold start scenario.

### 4.6. Performance in Different Parameters

4.6.1. Number of Neighbor Nodes

As shown in Figure 4, the AUC of DINK increases gradually with the increase in the number of item aggregations in MovieLens-1M and Last.FM datasets, but when the number of aggregations reaches 6, the AUC begins to retrace, while the AUC does not decline in the Book-Crossing dataset. This shows that in the case of sparse data, as the number of item sets increases, the features of aggregating items can play a positive role in recommendation, while in nonsparse cases they will play a negative role.

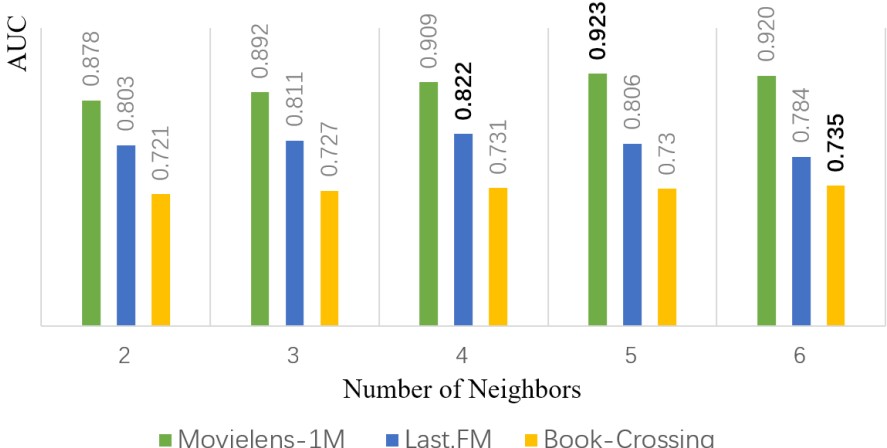

**Figure 4.** Performance in Different Number of Neighbors.

4.6.2. Length of Interaction Sequence

As shown in Figure 5, with other parameters unchanged, the AUC metrics of all three datasets keep rising as the length of the user interaction sequence increases. When the length of the user sequence reaches 12, the AUC indicator begins to flatten or even decline,

which indicates that the longer the user interaction sequence is, the better. When the length exceeds a threshold, DINK cannot extract more dynamic interest features.

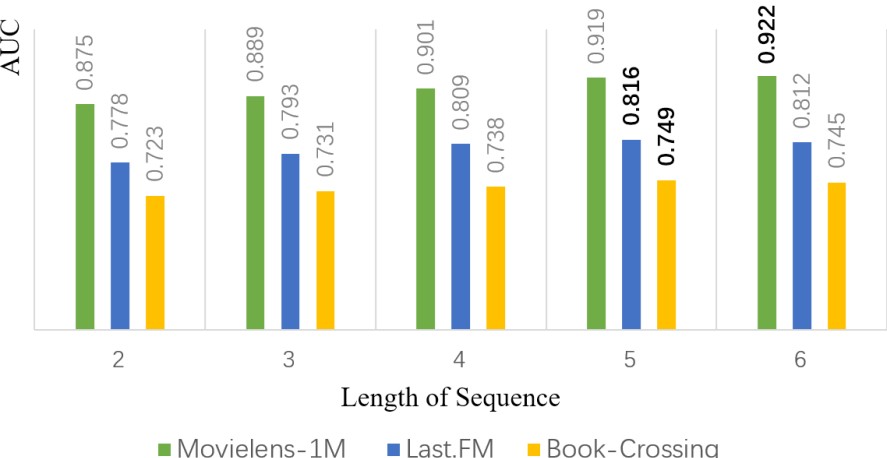

**Figure 5.** Performance in Different Lengths of Sequence.

4.6.3. Dimension of Embedding

As shown in Figure 6, with the other parameters unchanged, the AUC of DINK in MovieLens-1M will gradually increase with increasing d-dimension and achieve the optimal performance in the case of 64 dimensions. In the Last.FM and Book-Crossing datasets, the model achieves the best performance at 16 dimensions. However, as the dimension increases, the performance gradually degrades.

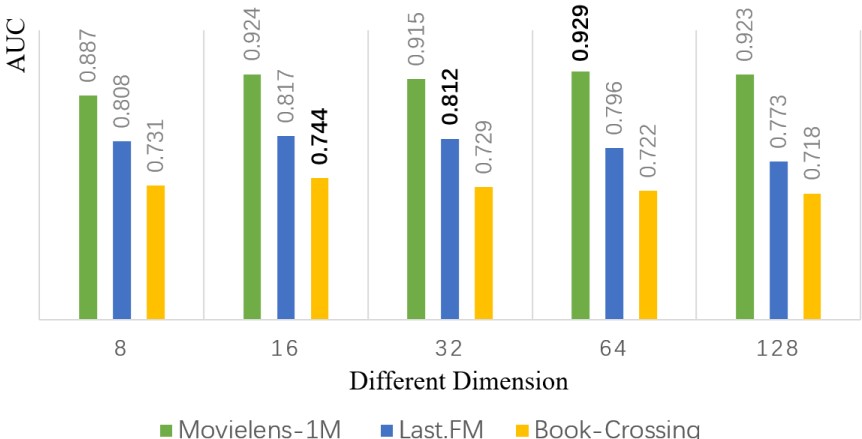

**Figure 6.** Performance in Different Dimensions.

**5. Conclusions**

Through research, we found that most recommendation models based on knowledge graphs take the static interaction matrix as the user's behavior and interest characteristics, ignoring the user's behavior and interest characteristics, which is a dynamic process. Aiming to resolve this problem, this paper proposes a deep interest network based on knowledge graph (DINK) to fuse users' dynamic interest features into recommendation tasks. DINK consists of three parts: an embedding layer, a dynamic interest reasoning layer and a prediction layer. In the embedding layer, the user interaction sequence is mapped into the recommended scene knowledge graph, and the features of the first-order neighbor nodes are extracted by the aggregator. In the dynamic interest reasoning layer, DINK calculates the similarity score between the user interaction sequence and the candidate item as the attention weight and then uses the weighted center aggregation vector as the input of the GRU to obtain the user's dynamic interest feature. In the prediction layer, the user

vector, candidate item vector and dynamic interest vector are spliced and pooled to predict the score of the candidate item. We conduct click-through rate prediction experiments, Top-k recommendation experiments, and performance experiments on three public datasets: Movielens-1M, Book-Crossing, and Last.FM. Experimental results show that, compared with other models, DINK can more effectively capture the dynamic interest features of users and can effectively alleviate the cold start problem.

**Author Contributions:** Conceptualization, D.Z. and H.W.; methodology, H.W., J.L., Y.M. and D.Z.; investigation, A.R. and X.Y.; visualization, H.W., A.R. and J.L.; project administration, D.Z.; writing—original draft preparation, H.W., D.Z., J.L. and A.R.; writing—review and editing, D.Z., H.W., Y.M. and A.R. All authors have read and agreed to the published version of the manuscript.

**Funding:** This research was funded by (i) Natural Science Foundation China (NSFC) under Grant No. 61402397, 61263043, 61562093 and 61663046; (ii) Open Foundation of Key Laboratory in Media Convergence of Yunnan Province under Grant No. 220225201. (iii) Practical innovation project of Yunnan University, Project No. 2021z34, No. 2021y128 and 2021y129.

**Acknowledgments:** This research was supported by the Yunnan Provincial Key Laboratory of Software Engineering, the Kunming Provincial Key Laboratory of Data Science and Intelligent Computing and the Key Laboratory in Media Convergence of Yunnan Province.

**Conflicts of Interest:** The authors declare no conflict of interest.

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
