# Peer review of "Deep Interest Network Based on Knowledge Graph Embedding"

_applsci, doi:10.3390/app13010357_

Round 1

Reviewer 1 Report

Authors in this work propose DINK, a knowledge graph-based deep interest exploration network, to extract users' dynamic interests. The authors claim that the DINK can be divided into a knowledge graph embedding layer, an interest exploration layer and a recommendation layer. Results are interesting. 

Author Response

Dear Reviewer, thank you for taking time out of your busy schedule to review our paper. We are very grateful for your recognition of our work and wish you a happy life.

Reviewer 2 Report

The paper presents a knowledge graph-based deep interest exploration network consisting of knowledge graph embedding layer, an interest exploration layer and a recommendation layer. The system aims at recommendations based on extracting users' dynamic interests. System's functionalities are demonstrated through a series of experiments on three public datasets.

The paper is well-structured, the methodology is scientifically sound, clearly presented and well-justified. The paper provides an overview of existing recommendation methods and systems and the critical analysis outlines the possible contributions of the proposed system.

I think it is worth discussing the considerable difference in the performance on different datasets. Considering the features of the datasets can explain some of the results. In general, a discussion on the comparison will be beneficial.

Some minor suggestions:

lines 19-20 restructure to avoid repetition, e.g.: Due to the diversity and heterogeneity information knowledge graphs represent, they are widely used...

lines 175-176 adjust sentence structure to make it clearer, e.g. maybe number the steps in sequence

line 234 I don't see the abbreviation KIEN introduced. Other abbreviations later on as well, e.g. in section 4.2

line 260 Use the same formatting for numbers, i.e., 17633 entities and 20195 relationships. > 17,633 entities and 20,195 relationships. Also in Table 1.

Author Response

Point 1: lines 19-20 restructure to avoid repetition, e.g.: Due to the diversity and heterogeneity information knowledge graphs represent, they are widely used...

Response 1: Thank you for your pertinent advice. We have taken this suggestion and made changes.

Point 2: lines 175-176 adjust sentence structure to make it clearer, e.g. maybe number the steps in sequence

Response 2: Thank you for your suggestion. The addition of serial numbers really makes the sentence more layered. We've made a change.

Point 3: line 234 I don't see the abbreviation KIEN introduced. Other abbreviations later on as well, e.g. in section 4.2

Response 3: We were really sorry for our careless mistakes. Thank you for your reminder. It has been modified. And we checked all the text to prevent the problem from happening again. Thanks again for your advice.

Point 4: line 260 Use the same formatting for numbers, i.e., 17633 entities and 20195 relationships. > 17,633 entities and 20,195 relationships. Also in Table 1.

Response 4: Thanks for your comments, we have modified the problem of inconsistent digital format, thank you for taking time out of your busy schedule to review our paper.

Reviewer 3 Report

Would you consider giving grand access to your code to reply to your results?

Author Response

Dear Reviewer, thank you for taking time out of your busy schedule to review our paper. We really understand your request to reproduce our code. However, this project is a private project of the laboratory, and we will have work to complete based on this project in the future, so please forgive us for not being able to provide the code.
